# Occult Hepatitis B Virus Infection: An Update

**DOI:** 10.3390/v14071504

**Published:** 2022-07-08

**Authors:** Carlo Saitta, Teresa Pollicino, Giovanni Raimondo

**Affiliations:** 1Division of Medicine and Hepatology, University Hospital of Messina, 98124 Messina, Italy; csaitta@unime.it; 2Department of Clinical and Experimental Medicine, University of Messina, 98124 Messina, Italy; 3Department of Human Pathology, University Hospital of Messina, 98124 Messina, Italy; tpollicino@unime.it

**Keywords:** covalently closed circular DNA, hepatitis B virus, HBV surface antigen, occult HBV infection

## Abstract

Occult hepatitis B virus (HBV) infection (OBI) refers to a condition in which replication-competent viral DNA is present in the liver (with detectable or undetectable HBV DNA in the serum) of individuals testing negative for the HBV surface antigen (HBsAg). In this peculiar phase of HBV infection, the covalently closed circular DNA (cccDNA) is in a low state of replication. Many advances have been made in clarifying the mechanisms involved in such a suppression of viral activity, which seems to be mainly related to the host’s immune control and epigenetic factors. OBI is diffused worldwide, but its prevalence is highly variable among patient populations. This depends on different geographic areas, risk factors for parenteral infections, and assays used for HBsAg and HBV DNA detection. OBI has an impact in several clinical contexts: (a) it can be transmitted, causing a classic form of hepatitis B, through blood transfusion or liver transplantation; (b) it may reactivate in the case of immunosuppression, leading to the possible development of even fulminant hepatitis; (c) it may accelerate the progression of chronic liver disease due to different causes toward cirrhosis; (d) it maintains the pro-oncogenic properties of the “overt” infection, favoring the development of hepatocellular carcinoma.

## 1. Introduction

The possible presence of hepatitis B virus (HBV) infection in subjects testing negative for the serum markers routinely used for diagnosing type B hepatitis has been suspected since the early 1970s, as reviewed in [1]. However, it was only in 1999 that the topic became of relevant and general scientific interest, when a milestone paper was published in the New England Journal of Medicine [2], showing the presence of HBV genomes in liver biopsy specimens from HBV surface antigen (HBsAg)-negative patients with chronic liver disease (CLD). This study revealed—for the first time—that subjects with CLD mainly due to hepatitis C virus (HCV) and occult HBV infection (OBI) were at a higher risk of progression toward cirrhosis than patients without OBI and that viral genetic heterogeneity was not responsible for a reduced viral replication or an impaired HBsAg synthesis. After that, a number of studies from many areas of the world have provided further, solid evidence concerning both the biology and the clinical implications of OBI. Giving the increasing interest in the topic, two international workshops were held in Taormina (Italy) in 2008 and 2018 [3,4], leading to the production and subsequent sharing of statements by a large number of attending international experts in the field. Here, we would like to review the existing knowledge on the virological and clinical aspects of OBI, highlighting both actual controversies and emerging perspectives.

## 2. Definition of OBI

OBI is defined as the presence of replication-competent HBV DNA in the liver—in the presence or absence of HBV DNA in the blood—of individuals testing negative for HBsAg assessed by currently available assays [4]. In this peculiar phase of chronic HBV infection (EASL GLs), HBV genomes are in the form of episomal covalently closed circular DNA (cccDNA) and in a low state of replication; thus, the detectability of HBV DNA in serum/plasma is intermittent, and when detectable, it is usually at a low viremic range, usually lower than 200 International Units (IU)/mL. As a consequence, the prevalence of detectable HBV DNA in serum is variable and depends on the population studied, the sensitivity of the assay used, and whether the blood samples analyzed were collected at one or more time points [5,6,7,8,9]. OBI can be classified as seropositive or seronegative according to the presence of serum markers of HBV exposure. Seropositive OBI refers to subjects with antibodies against the HBV core antigen (anti-HBc) and/or antibodies against HBsAg (anti-HBs) detectable in the serum and accounts for about 80% of all OBI cases [4]. Seronegative OBI refers to OBI individuals negative for all HBV serum markers including anti-HBc and anti-HBs, in whom the only detectable marker of HBV infection is the intrahepatic (and more rarely the circulating) HBV DNA. Seropositive OBI subjects may have lost HBsAg either after resolution of acute hepatitis B or after a chronic, long-lasting “overt” HBV infection; thus, the duration of HBsAg positivity before its disappearance may be highly variable. Individuals with seronegative OBI may have progressively lost anti-HBV antibodies, or they could be anti-HBV negative from the beginning of the infection. This latter primary seronegative occult infection has been described in the woodchuck model following inoculation with a low number (less than 100) of virions of hepadnavirus “woodchuck hepatitis virus” (WHV) [10]. In a small minority of OBI subjects, the absence of serum HBsAg may be related to an infection with HBV genetic variants carrying mutations at the S gene level, which leads to the production of modified HBsAg that is undetectable even by the sensitive commercially available assays. In these cases, serum HBV DNA levels can be as high as those commonly detected in subjects with “overt” HBsAg-positive infection [11,12,13,14,15,16,17,18].

Of note, HBV DNA may also be integrated into the host’s genome in individuals with OBI. However, integrated viral sequences are not involved in HBV replication, and their presence has no impacts on the making of an OBI diagnosis, since OBI typically occurs in cases with persistence of replication-competent HBV DNA [4].

## 3. Biology of OBI

The stability of the HBV cccDNA chromatinized episomes and their capability of long-lasting persistence into the nucleus of infected hepatocytes are the main molecular features of OBI. This stability, together with the long half-life of the hepatocytes, makes possible that HBV infection—once occurred—may last a lifetime even under conditions of the strong inhibition of viral functions [19,20,21,22,23,24,25,26,27,28]. The undetectability of HBsAg in OBI patients, despite the persistence of cccDNA, is generally thought to be due to the suppression of viral replication and gene expression exerted by epigenetic mechanisms and/or by the host’s immune control.

A number of studies have shown that some OBI patients carry a greater proportion of mutations in the pre-S/S region than patients with chronic hepatitis B (CHB), which may cause a reduction in antigenicity for the detection of HBsAg or an alteration in the production or secretion of HBsAg [29,30,31,32,33,34,35]. However, cccDNA in OBI cases is fully replication competent [36]. Moreover, the virus can be transmitted through blood transfusions and organ transplants [37,38], determining overt HBV infection in the recipients, and it may reactivate in patients undergoing immunosuppression [39,40,41]. Finally, OBI has been attributed to mutations inducing altered viral replication in very few cases [14,42]. Taken together, these considerations suggest that hosts rather than viral factors are more important in determining OBI occurrence. OBI is mainly associated with low concentrations of cccDNA in the hepatocyte nucleus, resulting in low amounts of HBV transcripts and protein expression and, consequently, in HBsAg undetectability [43]. The presence of transcriptionally silenced cccDNA suggests that gene expression can be regulated by epigenetic mechanisms in subjects with OBI. However, data on such epigenetic regulation of HBV transcription in OBI patients are few. One study showed that methylation patterns in HBV CpG islands were different in patients with occult and in those with overt chronic HBV infection, suggesting different epigenetic control mechanisms in patients with OBI and CHB [44]. In particular, OBI patients showed a higher methylation density in HBV CpG islands, a condition known to be associated with low viral replication activity in HBsAg-positive/HBeAg-negative patients [45]. A further mechanism that regulates HBV transcription is the post-translational modification of cccDNA-bound histones [46]. cccDNA is organized as nucleosomes, minichromosomal structures with histones, and other cellular or viral proteins [20]. Both in vitro and in vivo studies have shown that HBV transcriptional and replicational activities are regulated by the status of acetylation of cccDNA-bound histones (H3/H4) [24]. A considerable number of studies have shown that a number of cccDNA-linked proteins, such as the HBc and hepatitis B X (HBx) proteins, transcription factors, such as cAMP response element binding protein (CREB) and signal transcription factors 1 and 2, and many others are able to regulate HBV transcriptional activity [24,46,47,48,49,50,51,52,53]. However, even if all these epigenetic mechanisms may lead to a significative control of viral replication and, consequently, to very low HBV DNA levels and HBsAg undetectability, there is still a lack of direct evidence in OBI patients.

The role that the immune response to HBV plays in OBI is indirectly suggested by evidence of HBV reactivation in OBI patients undergoing immunosuppressive treatments or hemopoietic stem-cell transplantation [39,40,41]. However, the mechanisms involved in the immunological scenario present in OBI cases have not been sufficiently elucidated so far. In patients with spontaneously resolved acute HBV infection, a long-lasting HBV T-cell response leading to the control of HBV replication has been demonstrated [54,55]. Such HBV T-cell response has been observed at higher levels in OBI subjects than in individuals with HBsAg-positive CHB [29]. Another study showed different patterns of T-cell response against HBV epitopes in seropositive and seronegative OBI patients [56]. While ex vivo response was comparably weak in both groups, in vitro T-cell expansion after stimulation was more effective in seropositive than in seronegative OBI subjects [56]. Similarly, an acute WHV infection may resolve with the consequent long-lasting persistence of a seropositive occult infection [57,58]. As a note, in the woodchuck model of hepadnavirus infection, the exposure to very low, even if multiple, doses of WHV leads to seronegative occult WHV infection [10,59,60].

In addition to cellular immunity, antibody responses are also likely involved in the host’s control of OBI [41], as indicated by the frequent occurrence of HBV reactivation in patients treated with B-cell depleting monoclonal therapies, such us rituximab and ofatumumab [39,41,61]. Finally, innate immunity may also contribute to OBI control, as recently suggested by a study of the woodchuck model showing different expression of toll-like receptors in acute, chronic, and occult woodchuck hepatitis [62].

## 4. Diagnosis of OBI

The diagnosis of OBI relies on the detection of HBV DNA in the liver or in the blood of individuals testing negative for HBsAg. The gold standard is considered the detection of HBV genomes in liver DNA extracts. However, HBV DNA testing in the blood is a largely easier technique to be performed and a much more frequently used diagnostic approach. Above all, with the purpose to identify potential seropositive OBI subjects in cases of blood, tissue, or organ donation and when immune suppressive therapy has to be started, the use of anti-HBc testing may be adopted as a surrogate marker for OBI diagnosis [3,4]. Thus, a valid diagnosis of OBI is based on the sensitivity of the tests used to detect HBsAg and HBV DNA. The lower limit of detection (LLOD) of many commercially available HBsAg assays is 0.05 IU/mL, and it has been shown that up to 48% of negative samples tested with such assays turn positive using more sensitive HBsAg tests (LLOD of 0.005 IU/mL) [63,64,65,66,67]. Furthermore, anti-HBs probes directed against various epitopes of HBsAg should be mandatory for all HBsAg assays to enhance the detection of S-escape variants [4].

HBV DNA assays should perform consistently among HBV genotypes and subtypes, and the actual LLOD of many commercially available assays is 10–20 IU/mL. Considering that serum HBV DNA is frequently present at very low concentrations in OBI patients and that it can be intermittently revealed, the diagnosis of OBI requires the collection of blood samples at various time points and the testing of DNA extracts from at least 1 mL of serum or plasma [4].

In the setting of blood donations, highly specific (99.9%) and sensitive (LLOD 2–4 IU/mL) assays for nucleic acid testing (NAT) are used. However, when NAT screening is performed on minipools of multiple donations, sensitivity significantly decreases [68].

As mentioned above, the detection of replication-competent HBV DNA in the liver seems to be the ideal approach for diagnosis. However, no standardized assays with internal and external validation are yet available. The recommended approaches are nested-PCR techniques aimed at amplifying at least three different HBV genomic regions, real-time PCR techniques, or droplet digital PCR assays [2,4,21,69,70]. The proper processing of fresh (better than formalin-fixed) frozen liver samples is important to avoid cross-contamination, and appropriate negative and positive controls should be included to confirm specificity and validate the sensitivity of the assays [4].

## 5. Epidemiology of OBI

The worldwide epidemiology of OBI is quite variable, as it depends on many factors, such as the sensitivity of HBsAg and HBV DNA assays, whether there are risk factors for HBV exposure, the prevalence of HBV among the general population in the various geographic areas, the anti-HBV vaccination programs in the different countries, and the presence and severity of liver disease in the examined populations [71,72]. Indeed, most of the studies concerning OBI prevalence have been carried out on blood donors and on patients with liver disease; thus, they are not truly representative of the general population. Given that OBI prevalence is higher in areas of the world where hepatitis B is endemic [73], it should be noted that there are a few studies reporting low OBI prevalence in Asian and African areas with high endemicity of HBV infection [74,75,76,77]. High OBI prevalence has been found in groups of patients with risk factors for HBV infection, such as people who injected drugs (45%) [78], subjects with hepatitis C virus (HCV) co-infection (15–33%) [2,79] or human immunodeficiency virus (HIV) co-infection (10–45%) [80,81,82,83], and patients on dialysis (27%) [84]. Higher OBI prevalence rates have also been found in patients with co-existing liver disease, such as those with hepatocellular carcinoma (HCC) (63%) [21,85], cryptogenic cirrhosis (32%) [86], or liver-transplant patients (64%) [87]. Recently, two studies assessed OBI prevalence in patients with non-alcoholic fatty liver disease (NAFLD), one using anti-HBc as a surrogate marker and showing a prevalence of 15.5% [88] and one using the gold standard of HBV DNA detection in the liver tissues of morbidly obese patients undergoing bariatric surgery, which showed a prevalence of 12.8% [89]. One study investigated the presence of HBV DNA in the liver tissues of patients from different Italian regions and without liver disease who underwent abdominal surgery, showing a prevalence of OBI of 16% [90].

OBI is seldom found in blood donors. In well-conducted studies in this category of subjects, HBV DNA has been found in 0–4.6% of HBsAg-negative/anti-HBc-positive individuals, with a median prevalence of 1% [16,91,92,93,94,95].

## 6. Clinical Implications of OBI

The main reason behind the growing interest in OBI is related to accumulating evidence of its clinical impact. Indeed, (a) OBI can be transmitted, mainly through blood transfusion or liver transplantation, leading to hepatitis B in the recipient; (b) it can lead to viral reactivation in case of immunosuppression; (c) it may have a detrimental effect on the progression of CLD of different etiologies towards advanced clinical stages; (d) it has a significant role in the development of HCC (Figure 1).

### 6.1. Risk of OBI Transmission

#### 6.1.1. Blood Transfusion

Much evidence has shown that OBI carriers may transmit HBV infection through blood transfusion, with the consequent development of typical hepatitis B in the recipient [95,96,97,98,99]. Over the last 30 years, the risk of HBV transmission by blood transfusion has significantly decreased, thanks to the more sensitive diagnostic tools used for blood screening. However, the transmission of HBV from blood donors with OBI remains a major health problem in low- and middle-income countries, where anti-HBc tests and NAT are not widely diffused. A minimal risk of OBI transmission by transfusion also still exists in developed countries, because the minimum dose of infectious HBV DNA is below the LLOD of currently available NAT assays. If the donor is an OBI carrier, the transmission of the infection depends on many factors, such as the amount of plasma transfused, the immune status of the recipient, and the HBV serological status of both donor and recipient. Furthermore, considering that OBI is characterized by phases of transient viremia alternated with phases of absence of serum viral replication, an OBI carrier might be intermittently infectious. It has been shown that HBV-DNA-positive OBI donors with an isolated anti-HBc serological marker are more infectious than anti-HBs-positive OBI carriers [99] and that the anti-HBs positivity of the recipient significantly decreases the infection risk [100]. A recent study showed that three repeat HBsAg-negative blood donors with serum HBV DNA undetectable through highly sensitive NAT transmitted HBV infection to nine blood recipients [9]. This study reviewed the estimate of the minimum infective dose of HBV from the previous 20 IU/mL to about 3.0 IU/mL of HBV DNA [9]. Thus, the NAT sensitivity needed to prevent the transfusion-mediated transmission of HBV should be lowered from the actual 3.4 IU/mL to a new LLOD of 0.15 IU/mL [4].

#### 6.1.2. Liver Transplantation

The possibility of HBV transmission from a seropositive OBI liver donor to an HBV susceptible, seronegative recipient with the consequent development of hepatitis B is well known [37,101,102]. These recipients should receive long-term prophylactic antiviral therapy with nucleos(t)ide analogues (NUCs), such as entecavir or tenofovir. Nevertheless, even if a typical, “overt”, HBsAg-positive infection is prevented by NUC prophylaxis, it may not be able to prevent the development of OBI in the recipient [102,103]. Furthermore, patients who received a liver transplant for “overt” HBV infection may develop OBI of the liver graft even if they received antiviral prophylaxis; thus, lifelong NUC therapy is recommended in these patients [103,104,105]. Whether OBI might have an impact on the long-term outcome of transplanted patients is a matter of debate, even if evidence suggests that it can be responsible for a more rapid progression of liver disease following transplantation in HCV-positive patients [106].

### 6.2. OBI Reactivation

Chronic HBV-infected subjects undergoing immunosuppression are at high risk of viral reactivation, which may lead to a severe clinical outcome, even including fulminant hepatitis [107]. Such a reactivation may occur both in HBsAg-positive and in OBI individuals. Thus, the strong suppression of viral replication typical of the OBI status may cease in the case of immunosuppression because of the loss of immunological control of the virus activities, with consequent viral reactivation. This is a very important, although indirect, proof of the role played by immune response in inducing OBI status. The diagnosis of HBV reactivation in OBI patients can be made in the case of (a) HBsAg reappearance and/or at least 1 log increase above the LLOD of serum HBV DNA in a subject with previously undetectable HBsAg and serum HBV DNA and (b) at least 1 log increase of serum HBV DNA in a subject with previously detectable HBV DNA [4]. HBV reactivation can occur in up to 40% of OBI patients treated with immunosuppressive treatments and/or cancer chemotherapies, although the incidence is lower than that observed in HBsAg-positive patients. A higher risk (>10%) is seen in OBI patients treated with anti-CD20 containing regimens and myeloablative therapies for hematopoietic stem-cell transplantation [39,40,108,109,110,111]. A moderate (1–10%) or low (<1%) risk is seen in OBI patients treated with other anti-cancer therapies, high-dose corticosteroids, anti-TNF-α, or anti-rejection drugs used after solid organ transplant [8,112,113]. Recent studies have shown a low risk of reactivation in OBI patients treated with tumor necrosis factor inhibitors [114], as well as in those treated with direct-acting antiviral drugs for HCV infection [115,116]. Considering the immune deficiency caused by HIV, viral reactivation has been more frequently seen in OBI patients with acquired immunodeficiency syndrome [117]. However, giving the broad use of antiretroviral therapies, which include drugs with anti-HBV activity, the risk of HBV reactivation in HIV co-infected OBI patients has become negligible [118]. In the majority of the studies on HBV reactivation, the diagnosis of OBI has been based on the detection of anti-HBc. However, studies that have tested HBV DNA in the blood have shown a higher risk of viral reactivation in OBI patients with detectable HBV DNA, even if the risk is also still present in subjects with undetectable HBV DNA [92]. Viral reactivation may occur also in anti-HBs-/anti-HBc-positive patients, particularly in those in whom anti-HBs antibodies progressively decrease during immunosuppression [91,119,120]. The last European Association for the Study of the Liver (EASL) guidelines on the management of HBV infection suggest to treat with prophylactic antiviral therapy all HBsAg-negative/anti-HBc-positive patients undergoing high-risk immunosuppressive treatments and to continue such therapy for at least 18 months after the termination of immunosuppression (lamivudine can be used in this setting) [121]. They further suggest pre-emptive therapy in HBsAg-negative/anti-HBc-positive patients undergoing moderate- or low-risk immunosuppressive treatments. Practically, these patients should be frequently monitored and eventually treated with NUC in case of HBsAg reappearance and/or serum HBV DNA detection [121].

### 6.3. OBI and Chronic Liver Disease

A major and still widely debated topic is whether OBI may contribute to liver damage favoring the progression towards cirrhosis and the development of HCC in patients with CLD caused by different etiologic factors, such as HCV, non-alcoholic fatty liver, and alcohol-use disorders [122,123,124,125,126,127,128,129,130,131,132,133] (Table 1). Indeed, patients who recover from a self-limited HBV acute hepatitis may persistently carry HBV genomes for several decades without showing any clinical or biochemical sign of liver injury [122], but when their liver tissues are examined, the histological signs of mild hepatic inflammation are revealed [123,124]. These findings are in agreement with studies on the woodchuck model, showing the lifelong persistence of low amounts of replicating virus associated with a mild liver necroinflammation in woodchucks recovering from acute WHV infection [125,126].

Much evidence (including studies conducted in areas where HBV prevalence is low, such as the United States and Europe) indicates that OBI is a risk factor for the progression of liver disease towards cirrhosis in patients with HCV-related CLD [2,73,127,128,129]. Interestingly, some studies have shown an association between transaminase elevation and the reappearance of serum HBV DNA in chronic hepatitis C patients, suggesting a possible impact of such transitory HBV reactivation on hepatocyte necrosis [6,7]. A similar, negative impact of OBI on the progression of CLD has been shown in patients with alcoholic or cryptogenic liver disease and—recently—also in patients with NAFLD [72,88,89,130,131]. In particular, one study assessed the presence of HBV viral genomes in liver tissues collected from HBsAg-negative morbidly obese patients at the time of bariatric surgery, revealing a prevalence of about 13% of OBI, which was an independent predictor of NASH and fibrosis in these subjects [89].

Altogether, these data seem to confirm the hypothesis that—in conditions of immune competence—OBI is innocuous in itself, but when other causes of liver damage coexist (i.e., NAFLD, HCV infection, alcohol abuse), the minimal, prolonged state of inflammation provoked by OBI may contribute to the acceleration of the CLD towards the most advanced stages [133,134,135,136,137,138,139].

**Table 1 viruses-14-01504-t001:** Selection of reviews, meta-analyses, and statements concerning the relationship between occult hepatitis B virus infection and cirrhosis/hepatocellular carcinoma published since 2011.

Article	Type of Study
“Hepatocellular carcinoma: the point of view of the hepatitis B virus” Pollicino, T., et al., *Carcinogenesis* (2011) [140]	Review
“Association between occult hepatitis B infection and the risk of hepatocellular carcinoma: a meta-analysis” Shi, Y., et al., *Liver International* (2012) [141]	Meta-analysis
“Occult HBV infection” Raimondo, G., et al., *Seminars in Immunopathology* (2013) [71]	Review
“Occult hepatitis B virus and the risk for chronic liver disease: a meta-analysis” Covolo, L., et al., *Digestive and Liver Disease* (2013) [130]	Meta-analysis
“Occult hepatitis B virus and hepatocellular carcinoma” Pollicino, T., and Saitta, C., *World Journal of Gastroenterology* (2014) [142]	Review
“Occult hepatitis B virus infection and hepatocellular carcinoma: a systematic review” Huang, X., and Hollinger, F.B., *Journal of Viral Hepatitis* (2014) [134]	Review
“HBsAg-negative hepatitis B virus infection and hepatocellular carcinoma” Chen, L., et al., *Discovery Medicine* (2014) [135]	Review
“Occult hepatitis B virus infection” Kwak, M.S., et al., *World Journal of Hepatology* (2014) [136]	Review
“Update on occult hepatitis B virus infection” Makvandi, M., et al., *World Journal of Gastroenterology* (2016) [137]	Review
“Current knowledge of occult hepatitis B infection and clinical implications” Yip, T.C., et al., *Seminars in Liver Disease* (2019) [138]	Review
“Update of the statements on biology and clinical impact of occult hepatitis B virus infection” Raimondo, G., et al., *Journal of Hepatology* (2019) [4]	Statements
“Occult hepatitis B infection and hepatocellular carcinoma: Epidemiology, virology, hepatocarcinogenesis and clinical significance” Mak, L.Y., et al., *Journal of Hepatology* (2020) [139]	Review
“Occult hepatitis B virus infection in hepatitis C virus negative chronic liver diseases” Franzè, M.S., et al., *Liver International* (2022) [131]	Review

### 6.4. OBI and HCC

HBV is the main risk factor for HCC development worldwide; therefore, the World Health Organization has included it in “group 1” as a human carcinogen [140]. Many studies have revealed a strong association also between OBI and HCC. Retrospective and prospective studies including Asian or European patients with cryptogenic CLD and HCC have shown the presence of OBI in up to 70% of their liver tissues [21,43,143,144,145]. A number of studies comparing OBI prevalence between anti-HCV-negative patients with and those without HCC have revealed a significantly higher prevalence of the occult infection in HCC patients [21,143,146,147]. Analogously, many studies from Asia and Europe have shown that the prevalence of OBI is significantly higher in anti-HCV-positive patients with HCC than in those with CLD or in healthy controls [21,148,149,150,151]. An Egyptian study including anti-HCV-positive patients with HCC who underwent resection or liver transplantation showed that those with OBI (50% of the subjects analyzed) were younger and with a worse histological liver tumor grade [152]. Two observational cohort studies (with a median follow-up of 6.9 years and 11 years) analyzing the cumulative incidence of HCC in both cirrhotic and non-cirrhotic anti-HCV-positive subjects with and without OBI, showed that OBI carriers had a higher incidence of HCC than patients without OBI [128,153]. Finally, a metanalysis of 16 studies including 3256 patients showed an increased HCC risk in subjects with OBI [141].

Apart from the large number of above-mentioned association studies, OBI’s role in HCC development has been documented by a body of evidence showing that it maintains the same pro-oncogenic properties of “overt” infection. The hepatocarcinogenic activity of HBV is exerted through both direct and indirect pathogenetic mechanisms [142]. Concerning direct pro-oncogenic mechanisms, the capability of viral DNA to integrate into the host genome seems to play a major role. Some studies performed in the 1980s using hybridization technology have identified the presence of integrated viral DNA within the genome of HBsAg-negative individuals with HCC [154]. Subsequent studies based on more advanced molecular techniques, such as PCR-based and high-throughput sequencing assays, have confirmed the role of HBV integration in OBI-associated HCC [155,156,157,158,159]. Some of these studies have revealed the presence of HBV integrants in up to 75% of HCCs [145,157], a prevalence similar to that observed in HBsAg-positive HCC patients [160,161]. Furthermore, it has been shown that, analogously to what occurs in HBsAg-positive individuals, the portions of HBV viral genome integrated into the host genome often include the X and the pre-S/S genes both producing proteins with transforming properties, as well as viral regulatory elements, and that the host genes targeted by viral integration are very frequently involved in the cell cycle, proliferation, and immortalization [145,157]. Of note, many studies have shown that HBV integration may also be found in HBsAg-negative HCC patients without cirrhosis [145,157,162,163,164,165,166], further indicating the direct role of HBV DNA integration in HCC development also in OBI patients. Moreover, it has been shown that the cis-activation of cellular genes, such as *SERCA1* (the gene encoding sarco-/endoplasmic reticulum Ca^2+^-ATPase) and *PARD6G* (the partitioning-defective-6-homolog-gamma gene), can be a consequence of HBV integration in OBI non-cirrhotic patients with HCC [164,166]. The sites of HBV integration often show a loss of heterozygosity, deletions, duplications, and translocations [161]. Moreover, HBV integration has been found to be associated with replication errors and genomic instability [167], while conflicting results have been produced on the association between OBI, and mutations of *beta-catenin* and *p53* [162,163,168]. Finally, OBI patients show a high prevalence of pre-S2 variants in their HCC tissues [36,169], which are known to have hepatocarcinogenic potential in patients with chronic HBsAg-positive infection, since they induce the accumulation of mutated surface proteins in the endoplasmic reticulum (ER), leading to oxidative stress, DNA damage, and increased risk of HCC development [140].

## 7. Conclusions

OBI is a complex entity including a large spectrum of conditions widely divergent from each other from the virological and immunological points of view and also likely for their possible clinical impact.

The concept that HBV infection may persist lifelong in the hepatocytes even when viral functions are suppressed and the very large number of anti-HBc-positive individuals in vast areas of the world suggest that OBI is a frequent occurrence.

OBI is a very fascinating aspect of the viral hepatitis field both from the biological and clinical perspectives, and improvements in knowledge about it appear to be of extreme importance for a better comprehension of the epidemiology and the pathogenesis of HBV infection, which is still one of the major health problems in the world.

## Figures and Tables

**Figure 1 viruses-14-01504-f001:**
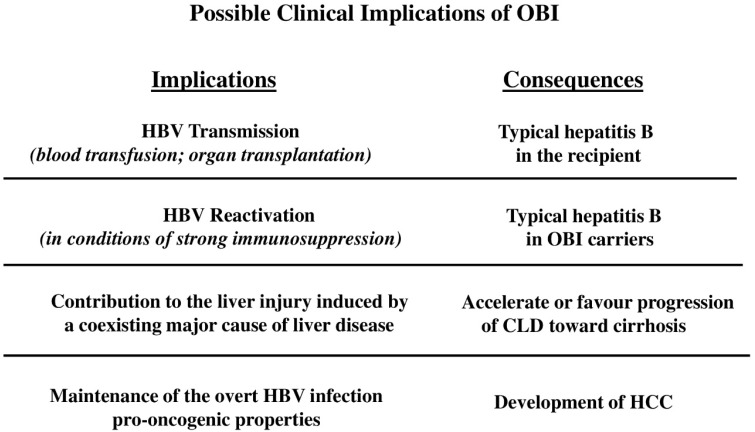
Schematic representation of the clinical impact of occult HBV infection (OBI).

## Data Availability

The data presented in this study are available in the included articles. No new data were created or analyzed in this study.

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
