# Peer review of "Occult Hepatitis B Virus Infection: An Update"

_viruses, 2022, doi:10.3390/v14071504_

Round 1

Reviewer 1 Report

The present review clearly summarizes recent knowledge about the biology and clinical implications of occult HBV infection (OBI), which remains a puzzling condition of the HBV infection natural history.

Minor comments:

- Page 2, line 66: to consider as “true”OBI HBV variants carrying HBsAg not detected by serological assays might be a matter of debate.  HBsAg variants not detected by the extremely sensitive assays currently used are now extremely rare.

- Page 2, line 91: “host rather than viral factors are more important in determining OBI occurrence”: this statement is also indirectly supported by reports showing that transfusion-transmission of OBI resulted in usual overt HBV infection in recipients.

- While agreeing with the authors that host factors are certainly the main driven force in the genesis of OBI, it may be mentioned that evidence of altered viral replication properties has been documented in some OBI-associated viruses. However, these are primarily case-specific and represent only a minority of OBI cases.

- Epidemiology section: data on the prevalence of OBI in general populations are lacking. As the authors point out, epidemiologic figures derived exclusively from patients and blood donors not truly representative of the general population.

- It might be interesting to discuss/mention the potential impact of HBV vaccination on OBI genesis.

- Few typing and formatting errors across the manuscript and especially in Table 1.

Reviewer 2 Report

Saitta et al wrote a comprehensive review about occult hepatitis B viral infection (OBI). Overall, this is a nice review, easy to read and very thorough. Main comments:

1) Regarding the risk of OBI reactivation during immunosuppressive therapy, Authors should discriminate the entity of risk according to the type of drug, e.g. antineoplastic agents (rituximab), therapy for autoimmune diseases (infliximab…) or steroids/antimetabolites.

2) Is there any link between OBI and HDV super- or co-infection?

3) Is there any evidence about direct anti-cccDNA antiviral agents for OBI?

4) Please give a short definition of potential OBI and relative implications for clinical practice (see Giannitti C et al, Clin Exp Rheumatol 2017).
